# Harnessing *γ-TMT* Genetic Variations and Haplotypes for Vitamin E Diversity in the Korean Rice Collection

**DOI:** 10.3390/antiox13020234

**Published:** 2024-02-14

**Authors:** Aueangporn Somsri, Sang-Ho Chu, Bhagwat Nawade, Chang-Yong Lee, Yong-Jin Park

**Affiliations:** 1Department of Plant Resources, College of Industrial Sciences, Kongju National University, Yesan 32439, Republic of Korea; aueangporn.somsri@gmail.com (A.S.); shchu@kongju.ac.kr (S.-H.C.); 26bhagwat@kongju.ac.kr (B.N.); 2Department of Industrial and Systems Engineering, College of Engineering, Kongju National University, Cheonan 31080, Republic of Korea; clee@kongju.ac.kr

**Keywords:** *Gamma-Tocopherol Methyl Transferase*, tocopherol, tocotrienol, vitamin E, diversity, haplotypes

## Abstract

*Gamma-tocopherol methyltransferase* (*γ-TMT*), a key gene in the vitamin E biosynthesis pathway, significantly influences the accumulation of tocochromanols, thereby determining rice nutritional quality. In our study, we analyzed the *γ-TMT* gene in 475 Korean rice accessions, uncovering 177 genetic variants, including 138 SNPs and 39 InDels. Notably, two functional SNPs, *tmt-E2-28,895,665-G/A* and *tmt-E4-28,896,689-A/G*, were identified, causing substitutions from valine to isoleucine and arginine to glycine, respectively, across 93 accessions. A positive Tajima’s *D* value in the *indica* group suggests a signature of balancing selection. Haplotype analysis revealed 27 haplotypes, with two shared between cultivated and wild accessions, seven specific to cultivated accessions, and 18 unique to wild types. Further, profiling of vitamin E isomers in 240 accessions and their association with haplotypes revealed that Hap_2, distinguished by an SNP in the 3′ UTR (*tmt-3UTR-28,897,360-T/A*) exhibited significantly lower *α*-tocopherol (AT), *α*-tocotrienol (AT3), total tocopherol, and total tocotrienol, but higher *γ*-tocopherol (GT) in the *japonica* group. Additionally, in the *indica* group, Hap_2 showed significantly higher AT, AT3, and total tocopherol, along with lower GT and *γ*-tocotrienol, compared to Hap_19, Hap_20, and Hap_21. Overall, this study highlights the genetic landscape of *γ-TMT* and provides a valuable genetic resource for haplotype-based breeding programs aimed at enhancing nutritional profiles.

## 1. Introduction

Tocochromanols, commonly known as vitamin E, are crucial for human nutrition due to their lipid-soluble antioxidant properties [1,2]. These amphiphilic lipids, comprising tocopherols and tocotrienols, play a vital role in human and animal diets. They are categorized into *α*-, *β*-, *γ*-, and *δ*-subforms based on the number and position of methyl groups on their structure [3,4]. Among them, *α*-tocopherol is prevalent in plant tissues like leaves, dicot seeds, and monocot embryos [5,6]. It is preferentially absorbed and retained in the human body, contributing significantly to its bioactivity in preventing various diseases such as inflammatory conditions, cardiovascular diseases, Alzheimer’s, cancer, and cataracts [7,8]. Tocotrienols, though rarer in nature than tocopherols, are gaining attention for potential anti-cancer, anti-inflammatory, and cholesterol-lowering properties. As research continues, the dietary and therapeutic significance of these compounds is ever-increasing [9].

The biosynthesis pathway of vitamin E in plants is a complex that primarily employs products from the shikimate metabolic (SK) and methylerythritol phosphate (MEP) pathways. Homogentisic acid (HGA), derived from the SK pathway, serves as a hydrophilic head, while phytyldiphosphate (PDP) or geranylgeranyl-diphosphate (GGDP) from the MEP pathway provides the hydrophobic tail, crafting the structure of tocopherols or tocotrienols [10]. This synthesis involves key enzymes like 4-hydroxyphenylpyruvic acid dioxygenase (HPPD/PSD1), Tocopherol cyclase (TC/VTE1), Homogentisate geranylgeranyl transferase (HGGT), Homogenesate phytyltransferase (HPT/VTE2), 2-methyl-6-phytylplastoquinol methyltransferase (MT/VTE3), and *γ*-tocopherol methyltransferase (*γ-TMT*/VTE4) [11,12]. Tocopherols are synthesized using PDP, whereas tocotrienols are produced with geranylgeranyl-diphosphate (GGPP). HGA phytyltransferases (HPTs) facilitate this reaction, displaying varying substrate preferences across different organisms. *Arabidopsis* HPT (*AtHPT*) preferentially uses PDP, but can also utilize GGPP to produce tocotrienols under specific conditions. This enzyme plays a role in the accumulation of diverse tocochromanol forms, influenced by HGA availability. Poaceae plants utilize HGGT for tocotrienol synthesis, accepting both PDP and GGPP as substrates. The products of these reactions are prenylated benzoquinols, serving as precursors for *δ*- and *β*-tocochromanols, or can undergo methylation to form dimethyl-benzoquinols, leading to *γ*-tocochromanols. The tocopherol cyclase (TC/VTE1) enzyme further cyclizes these compounds into *δ*- and *γ*-tocochromanols. Finally, the *γ*-tocopherol methyltransferase (*γ-TMT*/VTE4) converts *γ*- and *δ*-tocochromanols into *α*- and *β*-tocochromanols, respectively [13]. Given the crucial role of vitamin E, the tocopherol and tocotrienol pathway was unraveled in *Arabidopsis thaliana* L. [14], soybean (*Glycine max* L.) [15], tomato (*Solanum lycopersicum* L.) [16], and rice (*Oryza sativa* L.) [17].

Rice (*Oryza sativa* L.) stands as a pillar of sustenance for approximately half of the world’s population [18,19]. It thrives in diverse ecotypes, mirroring its adaptability to various environmental conditions and cultivation practices [20,21]. Over millennia, rice has diversified into several distinct ecotypes: the cold-resilient temperate *japonica*, grown primarily in temperate regions [22]; tropical *japonica* adapted to warmer climates [23]; *indica* with its long, slender grains ideal for tropical and subtropical zones [24]; *aus*, closely related to *indica* and named after its drought resistance [25]; and the *aromatic* group, distinguished for its unique fragrance, with basmati being the most renowned [22]. Additionally, there are *admixture* varieties, resulting from intermixing between primary rice groups, which encapsulate traits from multiple primary ecotypes [26]. Besides being a dietary staple, rice grains are rich in bioactive compounds, including vitamin E isoforms, flavonoids, phytic acid, *γ*-oryzanols, and phenolic compounds [27]. The *γ-TMT* gene is particularly significance in the context of vitamin E biosynthesis. It plays a pivotal role in modulating the metabolic flow of tocopherol synthesis, specifically catalyzing the conversion of *γ*-tocopherol to *α*-tocopherol [28]. Extensive research on *γ-TMT* underscores its significance. For instance, variations (such as insertions and deletions) in the *γ-TMT* gene in maize (*Zea mays* L.) have been shown to influence *α*-tocopherol levels critically [29]. Furthermore, extensive haplotype analysis has pinpointed beneficial alleles of *γ-TMT* to enhance vitamin E accumulation in maize [30]. In rice, analysis of 34 accessions revealed considerable diversity in vitamin E isomers across rice cultivars [31]. Moreover, a genome-wide association study (GWAS) identified *γ-TMT* as a major candidate gene influencing vitamin E variation in 137 rice accession references. Studies in other plants, such as Arabidopsis, soybean, and maize, have revealed that overexpressing the *γ-TMT* gene can lead to a substantial increase in tocochromanol accumulation [15,32,33]. These findings emphasize the transformative potential of *γ-TMT* in optimizing the pathway to boost vitamin E levels in crops, promising enhanced nutritional benefits.

With the advent of next-generation sequencing (NGS), understanding the genetic architecture of rice has been revolutionized. This study investigates the genetic architecture of the *γ-TMT* gene across 475 rice accessions, aiming to uncover functional haplotypes, natural genetic variations, and the evolutionary dynamics among ecotypes. Our findings will provide invaluable insights for rice breeding, potentially enhancing its nutritional profile.

## 2. Materials and Methods

### 2.1. Plant Materials

In this study, we utilized the Korean rice (KRICE) collection, comprising 475 accessions, including 421 cultivated and 54 wild accessions (Appendix A). The core set of 421 rice accessions was previously selected from a global collection of 4406 accessions by the National Genebank of the Rural Development Administration (RDA-Genebank, Jeonju-si, Republic of Korea) using the PowerCore program [34]. Based on whole-genome resequencing data, these accessions were categorized into six ecotypes: temperate *japonica* (*n* = 279), tropical *japonica* (*n* = 26), *indica* (*n* = 102), *aus* (*n* = 9), *aromatic* (*n* = 2), *admixture* (*n* = 3) [35]. Additionally, we included 54 wild rice accessions sourced from the International Rice Research Institute (IRRI) to provide a broader perspective on the genetic background.

### 2.2. Resequencing Data Processing and Variant Calling for γ-TMT Gene

To gain insights into the evolution of the *γ-TMT* gene within the KRICE collection, we analyzed resequencing data obtained from 475 rice accessions using a HiSeq 2500 sequencer (Illumina, San Diego, CA, USA), with an average coverage of approximately 15× [36]. The resequencing data underwent a series of processing steps, including data preparation, filtering, mapping, sorting, and variant calling. Firstly, we processed the raw data to ensure an average quality score (QS) of at least 20 per read using SICKLE (https://github.com/najoshi/sickle accessed on 1 August 2023) for trimming the 3′-end of reads. The decoded sequences were saved in fasta Q file format. To eliminate missing values, we used VCFtools (variant call format) version 0.1.15 [37], filtering each SNP for a ‘maximum missing’ value of 0.8 (allowing a maximum of 20% missing data) and a minor allele frequency (MAF) of 0.05. Subsequently, high-quality reads were aligned to the reference genome IRGSP-1.0 sequence using BWA v0.7.15, and Samtools v1.3.1. The aligned reads were stored in BAM (binary alignment map), and PICARDv1.88 was used to filter duplicate reads (Toolkit 2019). Subsequently, variant calling was performed using the Genome Analysis Tool Kit-V4.0.1.2 (GATK) [38] and filtered with VCFtools-V0.1.15 to eliminate potential false-positive SNPs/InDels, retaining variants with a minor allele frequency (MAF) of 0.05 or higher and a maximum missing data ratio (MDR) of 0.2. Finally, the *γ-TMT* gene region (chromosome 2, Chr02_28,894,065…..28,897,418) was extracted using BCFtools (v1.8) to retrieve the genetic variants within the gene and utilized for subsequent analyses.

### 2.3. Population Structure Analysis

In our study, we investigated the population structure of KRICE in relation to the *γ-TMT* gene. For this purpose, we utilized PLINK v1.07 [39] to transform the VCF file into a plink format. Using a custom Python script, we generated both bim and fam files. The population structure was then evaluated with fastStructure [40], where we tested K-values from 2 to 7. Subsequently, we employed the average Q-values in POPHELPER v2.3.1 to infer *admixture* patterns [41]. For a visual representation of the *γ-TMT* genetic composition within KRICE, we conducted a principal component analysis (PCA) using TASSEL5 [39]. The first two principal components, PC1 and PC2, were illustrated as a two-dimensional (2D) scatterplot generated with ggplot2 v3.6.2.

### 2.4. Nucleotide Diversity, Tajima’s D Analysis, and Fixation Index (F_ST_)

To gauge the polymorphism among the ecotypes of KRICE, we evaluated nucleotide diversity (π) and derived Tajima’s *D* values. We utilized VCFtools to extract the *γ-TMT* gene region for each respective group. A sliding window size of 500 bp was adopted for both Tajima’s *D* and nucleotide diversity analyses, and subsequent values were contrasted across ecotypes. Moreover, the fixation index (F*_ST_*) was estimated to ascertain genetic disparities across groups concerning the *γ-TMT* gene. This involved using a 500 bp sliding window set at 500 bp intervals.

### 2.5. Haplotype Analysis

We imported the VCF file encompassing the *γ-TMT* gene region from KRICE into TASSEL 5.0 to determine genetic variants across the accessions. To perform the haplotype analysis, this VCF file was transformed into FASTA format and subsequently aligned using MEGAX [42]. The alignment, after conversion to nexus format, was loaded into DnaSP (v6.12.03) [43] to ascertain the total number of haplotypes. A catalog of mutated sequences was curated for each group, laying the foundation for constructing the haplotype network utilizing Population Analysis with Articulate Tree (PopARTv1.7) [44] to form a TSC network. The dendrogram was constructed using FigTree version 1.4.3 and was based on the neighbor-joining method, encompassing 1000 bootstrap replications. This dendrogram was further visualized with the online tool iTOL [45].

### 2.6. Determination of Vitamin E Isomers by Gas Chromatography (GC)

The preparation of rice samples was adapted from the method presented by [46], with minor adjustments. Firstly, rice grains were de-hulled and finely ground. A mixture was created using five grams of the resulting rice powder and 20 mL of absolute ethanol. Subsequently, 0.5 g of ascorbic acid, serving as an antioxidant agent, was incorporated into each sample tube. These tubes were then submerged in an 80 °C water bath, agitated at 150 rpm for 10 min. To this mixture, 600 µL of 44% KOH was added, and the heating step was reiterated for 18 min. Once the saponification was complete, the samples were promptly cooled in an ice bath for 30 min. This was followed by the addition of 10 mL of distilled water and 10 mL of n-hexane. After centrifugation at 3000 rpm and 4 °C for 5 min, the supernatant was transferred to a new tube. This extraction step was performed thrice.

For purification, about 10 mL of distilled water was added, and the samples were passed through anhydrous sodium sulfate (Na_2_SO_4_) to remove any residual water. Afterward, samples were concentrated using a rotary evaporator (EYELA). For dissolution, 1 mL of iso-octane was added to each extracted sample tube. A total of 1 mL of this solution was transferred into a GC vial, from which 1 µL was injected into the GC instrument (GC-450, Variant). The instrument was equipped with a flame ionization detector and a CP-SIL 8CB column (30 m × 0.25 mm, 0.4 µm film thickness, Agilent, Santa Clara, CA, USA) set at 290 °C, operated with a split ratio of 1:20 and a continuous helium gas flow (1.0 mL/min). The initial temperature was set at 220 °C for 2 min, raised to 290 °C at a rate of 5 °C/min, maintained for 14 min, increased to 310 °C at 10 °C/min, and then held for 18 min. All analyses were performed in triplicates, with data from three biological replicates.

Compounds including *α*-tocopherol, *γ*-tocopherol, *α*-tocotrienol, and *γ*-tocotrienol were quantified based on standard curves. To create the standard curves, a series of known concentrations of eight tocochromanol isomers obtained from Sigma Aldrich (tocopherols: *α*, *β*, *γ*, *δ*, and tocotrienols: *α*, *β*, *γ*, *δ*) were subjected to the same GC analysis. The resulting detector responses for these known concentrations were plotted to establish a linear relationship. Using these standard curves, the concentrations of these compounds in unknown samples were determined (Appendix A).

### 2.7. Statistical Analysis

We utilized R (v4.2.3) for all statistical evaluations. To probe the relationship between functional *γ-TMT* haplotypes and variations in tocochromanol, a haplotype–trait association analysis was conducted. For accessions carrying similar haplotypes, the average tocochromanol concentration was determined and Student’s *t*-test was then employed to determine statistical differences in tocochromanol concentrations between the haplotypes.

## 3. Results

### 3.1. Genetic Polymorphisms in the γ-TMT of KRICE

Genetic variants within the *γ-TMT* gene of 475 KRICE samples were categorized into three main types: single-nucleotide polymorphisms (SNPs), insertions (Ins), and deletions (Dels). Of the 177 polymorphic sites detected in KRICE, 122 resided in intronic regions (Figure 1A). Within the transcript regions, 44 of the 55 identified sites were situated in exons, and 11 polymorphisms were observed in the 3′UTR region (Figure 1A). The wild group demonstrated the most pronounced genetic variants. Specifically, *O. nivara* and *O. rufipogon*, each with three accessions, exhibited 21 and 20 variations, respectively, whereas the remaining forty-eight wild accessions presented 140 variations. Among the cultivated varieties, *indica* had the highest genetic variants, trailed by *admixture* and then *aus* (Figure 1B).

### 3.2. Population Structure of KRICE Based on the γ-TMT 

To elucidate the composition of KRICE in terms of shared genetic variation in *γ-TMT*, we conducted a population structure analysis. Evaluating varying K values, ranging from 2 to 7, we observed mixed groupings among the samples (Figure 2A). Notably, at K = 3, the *indica* and *japonica* groups separated, while the wild group displayed a mixed pattern. Tropical *japonica* mirrored the structure of temperate *japonica*, and *aus* aligned closely with *indica* across all K values. With increasing K values, fragmentation became more pronounced within *indica* and wild accessions. Distinct genetic architectures for *indica* and *aus* were evident at K = 6 and K = 7, underscoring potential genetic divergence within the *γ-TMT* gene region for these groups. In contrast, throughout all K values, wild accessions consistently demonstrated an *admixture* pattern, signifying genetic overlap and allele sharing between wild and cultivated groups (Figure 2A).

We further investigated the principal component analysis (PCA) to ascertain the genetic relationships within the *γ-TMT* gene region. The PCA results revealed three distinct clusters, as depicted in Figure 2B. Cluster A encompassed all *japonica*, wild, *aromatic*, and some *indica* accessions. Cluster B comprised *aus* and *indica* accessions, while Cluster C was exclusively populated by *indica* accessions. The genetic distribution was more dispersed among the wild and *indica* groups, with a majority of wild accessions falling into Cluster A (Figure 2B). Collectively, these findings suggest that the wild subgroup shares a closer genetic similarity with cultivated groups, especially the *japonica* varieties.

### 3.3. Assessment of Population Differentiation via γ-TMT Gene Variations

To estimate the degree of population differentiation stemming from variations in the *γ-TMT* gene, we utilized the fixation index (F*_ST_*) as an indicator (Figure 2C and Appendix A). The highest F*_ST_* values observed were 0.08824 (between temperate *japonica* and *admixture*), followed by 0.8357 (temperate *japonica* vs. *aus*) and 0.7120 (temperate *japonica* vs. *indica*). A high level of gene migration (low F*_ST_*) was noted between *admixture* vs. *aromatic* and wild vs. *aromatic* groups, with F*_ST_* value = 0, followed by *admixture* and *indica* groups (0.0249), indicating relatively low differentiation among these groups. Interestingly, there was evidence of substantial gene flow (indicated by low F*_ST_* values) between the *admixture* and *aromatic* groups, as well as between the wild and *aromatic* groups, both registering an F*_ST_* value of 0. Similarly, a modest F*_ST_* value of 0.0249 was observed between the *admixture* and *indica* groups, further emphasizing their close genetic alignment. Notably, the wild group exhibited lesser differentiation compared to tropical *japonica* (0.2311), temperate *japonica* (0.4727), and *indica* (0.4348) than the differentiation observed between *indica* vs. tropical *japonica* (0.5347) and *indica* vs. temperate *japonica* (0.7120) (Figure 2C).

### 3.4. Polymorphism Assessment in KRICE Groups Based on the γ-TMT Gene

To assess the extent of polymorphism influenced by the *γ-TMT* variations across KRICE groups, we computed nucleotide diversity (π) values. Among the groups, the *indica* group displayed the highest mean nucleotide diversity (π = 0.0060), followed by wild (π = 0.0045), tropical *japonica* (π = 0.0015), and temperate *japonica* (π = 0.0009) (Figure 3A). In a sliding window analysis, the highest π value for the *indica* group was recorded at position 2,889,450, registering at π = 0.00552. Conversely, peak values for temperate *japonica* and tropical *japonica* were substantially lower, registering at 0.00022 and 0.00028, respectively. Towards the terminal region of the *γ-TMT* gene, there was a notable decline in π values for the *indica* and *japonica* groups. However, the wild group’s π values displayed an upward trend (Figure 3B).

### 3.5. Insights into Selection Patterns within the γ-TMT Gene

To further understand the differences between observed nucleotide diversity and the estimated number of segregating sites for the *γ-TMT* gene, we applied Tajima’s *D* test. The average Tajima’s *D* value for the *indica* group (0.6293) was significantly higher than that of the *japonica* and wild groups (Figure 3C). The *indica* group recorded the highest Tajima’s *D* value of 2.09848 at position 28,897,500. This positive value suggests an excess of intermediate-frequency alleles in the *indica* group, possibly indicative of balancing selection or a population contraction (Figure 3D). On the other hand, both *japonica* groups (temperate and tropical) and the wild group consistently displayed negative Tajima’s *D* values across all positions, indicating an abundance of rare alleles. This can be seen as evidence for either recent population expansion or positive selection in these groups.

### 3.6. Functional Polymorphisms in the γ-TMT and Phylogenetic Analysis

Within the transcript region of the *γ-TMT* gene, we identified forty-six SNPs and nine InDels (Appendix A). Among these, 28 polymorphic sites, including nonsynonymous SNPs and indels in exons, caused amino acid changes (hereafter referred to as fSNPs). Notably, among 55 polymorphic sites, 48 were present in wild accessions, while only nine were identified in cultivated accessions (Figure 4). In cultivated rice, two nonsynonymous SNPs, including G/A substitution in exon 2 (*tmt-E2-28,895,665-G/A*), caused a valine (V)-to-isoleucine (I) change, while another nonsynonymous A/G substitution in exon 4 (*tmt-E4-28,896,689-A/G*) resulted in an arginine (R)-to-glycine (G) change. Interestingly, both these alleles were detected in a subset of ninety-three accessions, which comprised eighty-seven *indica*, four *aus*, and two *admixture* accessions. In contrast, the wild-specific fSNPs, consisting of seventeen SNPs and eight indels, were present in 26 wild accessions (Appendix A).

To gain deeper insights into the evolutionary relationship based on these polymorphisms, we constructed a phylogenetic tree using the maximum likelihood method. This analysis revealed that KRICE accessions could be grouped into three main clusters. The major clade, Cluster-1, predominantly comprised cultivated rice accessions but also included some wild accessions, hinting at their close genetic lineage (Figure 5). Interestingly, cultivated accessions carrying the two aforementioned fSNPs, *tmt-E2-28,895,665-G/A,* and *tmt-E4-28,896,689-A/G*, formed Cluster-2. In contrast, Cluster-3 was predominantly populated by wild accessions (Figure 5).

### 3.7. Haplotype Diversity in the γ-TMT Gene Region

The genetic variants found in the *γ-TMT* were categorized into 27 distinct haplotypes. Notably, only two haplotypes, Hap_1 and Hap_2, were common to both cultivated and wild rice groups. Seven haplotypes (Hap_19 to Hap_25) were unique to cultivated accessions, whereas the remaining eighteen haplotypes (Hap_3 to Hap_18, Hap_26, and Hap_27) were specific to the wild group (Figure 4 and Appendix A). Hap_1, sharing its sequence with the Nipponbare reference, was the most prevalent, encompassing 278 accessions. This group included 251 accessions from temperate *japonica*, 23 from wild, 2 from tropical *japonica*, and 1 from *indica* (Figure 4). Hap_2, distinguished by an SNP in the 3′ UTR (tmt-3UTR- 28,897,360-T/A), was present in 73 accessions across all groups, including tropical *japonica* (24), temperate *japonica* (25), *indica* (13), *aus* (5), *aromatic* (1), *admixture* (1), and wild (4). Furthermore, seven haplotypes specific to cultivated accessions were characterized by seven SNPs and two indels (Figure 4). Hap_19, harboring two fSNPs, *tmt-E2-28,895,665-G/A*, and *tmt-E4-28,896,689-A/G*, was found in 21 *indica* accessions. In addition to the above two fSNPs, *tmt-3UTR-28,897,360-T/A* formed Hap_20 with 12 accessions. Two SNPs from the 3′ UTR along with *tmt-E2-28,895,665-G/A*, and *tmt-E4-28,896,689-A/G* detected in 56 accessions (55 *indica* and 1 *aus*) were grouped in Hap_21. Interestingly, Hap_24, characterized by a 1 bp deletion at 28,897,359, was found in three temperate *japonica* accessions. Additionally, haplotypes specific to wild accessions were characterized by twenty-two SNPs, three deletions, and five insertions (Figure 4).

To unravel the genetic relationships within KRICE ecotypes, we constructed a haplotype network. Our analysis spotlighted a total of seven haplotypes that branched directly from the predominant Hap_1. Interestingly, six of these were exclusive to the wild types. The cultivated-specific haplotype, Hap_20, appears to have evolved through a few mutational steps from Hap_1. Remarkably, the wild-specific Hap_15, Hap_16, Hap_25, and Hap_27 appeared to have been derived from Hap_2. Furthermore, Hap_8, which branched from Hap_1, led to the emergence of another five distinct haplotypes (Figure 6).

### 3.8. Phenotypic Effect of TMT Haplotypes on Tocochromanol Accumulation 

We explored the association between *γ-TMT* haplotypes and tocochromanol concentrations in cultivated rice. The analyzed haplotypes included Hap_1 (138 accessions), Hap_2 (55 accessions), Hap_19 (9 accessions), Hap_20 (9 accessions), and Hap_21 (9 accessions), totaling 240 selected accessions (Appendix A). Significant differences in the levels of *α*-tocopherol (AT), *α*-tocotrienol (AT3), total tocopherol (TT), and total tocotrienol (TT3) were observed across these haplotypes. Hap_1 displayed notably higher levels of these compounds, except for total tocotrienol, compared to Hap_19. Conversely, *γ*-tocopherol (GT) and *γ*-tocotrienol (GT3) contents were lower in Hap_1, with the exception of GT3 recorded between Hap_1 and Hap_2 (Figure 7). Hap_2 also exhibited higher AT, AT3, and total tocopherol levels, but lower GT and GT3 compared to Hap_19, Hap_20, and Hap_21. The latter three haplotypes showed no significant differences among themselves, except for AT3 between Hap_20 and Hap_21, and GT3 between Hap_19 and Hap_20 (Figure 7). Total vitamin content was significantly higher in Hap_1 and Hap_2 compared to other haplotypes (Appendix A). When comparing ecotypes, a significant decrease in AT, AT3, TT, and TT3 was observed in Hap_2 relative to Hap_1 among *japonica* accessions (Appendix A). In *indica* varieties, AT and AT3 levels were significantly lower in Hap_19 to Hap_21, while GT and *γ*-tocotrienol GT3 levels increased. However, TT and TT3 showed no significant differences, except between Hap_2 and Hap_19 for TT3 (Appendix A). In the *aus* group, accessions of Hap_2 and Hap_20 displayed significant differences in AT and AT3 levels (Appendix A). The total vitamin content did not show any significant variations in *indica* haplotypes, but in *japonica* and *aus* groups, a significant decrease was observed between Hap_1 and Hap_2, and between Hap_2 and Hap_20, respectively (Appendix A).

## 4. Discussion

The growing health consciousness and demand for nutritionally enriched foods have heightened interest in the nutritional quality of rice, particularly in relation to consumer acceptance. This trend has spurred the development of elite rice varieties enriched with bioactive compounds, especially focusing on enhancing vitamin E content [27]. Nonetheless, it is crucial to recognize that the vitamin E content in rice is profoundly influenced by both genetic variations among cultivars and environmental factors [47]. Given that vitamin E deficiency is linked to a variety of serious health issues worldwide [8,48] and humans cannot naturally synthesize this nutrient, dietary intake becomes essential [49]. 

The *γ-TMT* gene has emerged as a focal point in metabolic engineering efforts aimed at increasing *α*-tocopherol levels in crops [50]. In rice, two main subspecies are preferred in Asia: *indica*, predominantly grown in tropical and subtropical environments, and *japonica*, widely cultivated in temperate climates. These varieties display distinct divergences in morphological characteristics, agronomic traits, environmental adaptation, and physiological and biochemical properties through the domestication process [51]. Consequently, exploring genetic variations for micronutrient content in crops like rice is a vital strategy for improving the nutritional quality of human diets on a large scale [52]. In rice, the *γ-TMT* gene plays a pivotal role in vitamin E accumulation, particularly in *japonica* and *indica* varieties [7]. A comprehensive understanding of their evolutionary background and genetic diversity is key to deploying this knowledge for targeted improvements in rice breeding programs.

In our study, we investigated the genetic variation within the *γ-TMT* gene region, utilizing whole-genome resequencing data from a core set of 475 accessions. This approach facilitated our exploration of genetic diversity, evolutionary patterns, and haplotypes in the *γ-TMT* gene. Variant analysis of the *γ-TMT* gene revealed 177 genetic variants, comprising 138 SNPs and 39 InDels, with the highest level of genetic diversity observed in wild rice compared to its cultivated counterparts (Figure 1). In our population structure analysis, we observed a distinct separation of cultivated groups from wild types at most K-values, particularly from K = 5 to 7, indicating distinct cultivated groups (Figure 2A). Further PCA analysis revealed an admixture pattern in wild accessions, suggesting genetic overlap and allele sharing between wild and cultivated groups (Figure 2B). Additionally, genetic differentiation, assessed using the fixation index (F*_ST_*), showed high F*_ST_* values between *indica* and *japonica* groups (both tropical and temperate), implying significant genetic differentiation between these two subspecies (Figure 2C). The concept of nucleotide diversity, essential in molecular genetics, measures the degree of polymorphism within a population [50]. Our findings indicate a reduction in genetic diversity during domestication, more pronounced in the *japonica* group than in *indica*. This is evidenced by the lower nucleotide diversity in *japonica* following *indica* (Figure 3). Previous research has highlighted a narrower genetic diversity in *japonica* compared to *indica* rice, likely due to a domestication bottleneck [22,53]. Furthermore, breeding efforts over the past century have further narrowed the genetic diversity among cultivars, particularly those selected for cultivation in diverse agro-ecological environments [53,54]. In terms of Tajima’s *D* measurement, *indica* was the only group with a positive value, suggesting that rare alleles are more likely to be lost during bottleneck events [55]. Conversely, the occurrence of negative values in other groups, especially in wild rice, indicates an accumulation of segregation sites at rare frequencies, consistent with population expansion scenarios, such as selective sweeps or post-bottleneck expansion [55,56,57].

Analyzing haplotype variation offers a promising approach to identify different types of polymorphisms located on the same chromosome. These polymorphisms tend to be inherited together with minimal alteration due to contemporary recombination [58]. In our study, 27 distinct haplotypes were identified within the transcript region of all the rice accessions. Notably, the predominant haplotype, Hap_1, was shared between both cultivated and wild types, encompassing 278 accessions. Similarly, Hap_2, characterized by the SNP *tmt-3UTR-28,897,360-T/A*, was found in both wild and cultivated accessions. Additionally, two functional SNPs (fSNPs), *tmt-E2-28,895,665-G/A* and *tmt-E4-28,896,689-A/G*, leading to amino acid substitutions from valine (V) to isoleucine (I) and arginine (R) to glycine (G), respectively, were detected in 93 accessions. These accessions were distributed across five haplotypes due to variations in the 3′UTR (Figure 4). Interestingly, the fSNP *tmt-E2-28,895,665-G/A* was also identified in a genome-wide association study (GWAS) involving 137 accessions, where it was associated with AT content [7]. Among haplotypes, Hap_19, Hap_22, and Hap_24 were specific to *indica* varieties, while Hap_20 (comprising seven *indica*, four *aus*, and one *admixture*) and Hap_21 (fifty-five *indica* and one *admixture*) were predominantly found in *indica* accessions (Figure 4). To further explore the *γ*-genetic variants in a larger set of accessions, we queried the RiceVarMapv2.0 database (http://ricevarmap.ncpgr.cn/ accessed on 20 September 2023) [59]. Among a dataset of 4726 accessions, a total of 80 polymorphisms were detected, of which 59 were shared with KRICE. The A and G alleles of *tmt-E2-28,895,665-G/A* and *tmt-E4-28,896,689-A/G*, respectively, showed a frequency of 53.1% in all accessions, 85% in all *indica*, 92.3% in *indica*-I, 82.9 in *indica*-II, 49.4% in *aus*, and only 0.03% in tropical *japonica*, and 0.06% in temperate *japonica* (Appendix A). This distribution underscores the genetic diversity and specificity of haplotypes in relation to rice ecotypes and their potential implications for rice breeding and improvement.

In our study, a detailed pairwise comparison among haplotypes revealed significant differences in tocochromanol content. Notably, the reference haplotype Hap_1 displayed significantly higher levels of *α*-tocopherol (AT), *α*-tocotrienol (AT3), and total tocopherol (TT) compared to other haplotypes, as determined by a pairwise t-test at a significance level of 0.005. However, *γ*-tocopherol (GT) and *γ*-tocotrienol (GT3) levels were notably lower in Hap_1, except GT3 when comparing Hap_1 to Hap_2 (Figure 7). Accessions with Hap_2, distinguished by the SNP *tmt-3UTR-28,897,360-T/A*, showed significant differences in all compounds compared to Hap_20, which harbors the same *tmt-3UTR-28,897,360-T/A* and two fSNPs (*tmt-E2-28,895,665-G/A* and *tmt-E4-28,896,689-A/G*) (Figure 7 and Appendix A). In the *japonica* group, Hap_2 was associated with significantly lower AT, AT3, TT, TT3, and total vitamin E content, but higher GT (Appendix A), while *indica* haplotypes, Hap_19 with two fSNPs (*tmt-E2-28,895,665-G/A* and *tmt-E4-28,896,689-A/G*), Hap-20, and Hap_21 (which possess *tmt-3UTR-28,897,309-C/T and tmt-3UTR-28,897,387-T/A*, respectively, in addition to two fSNPs), displayed no significant differences among them (Appendix A). UTRs, particularly the 3′ UTR, are known to contain elements that influence the stability, localization, and translation efficiency of mRNAs [60]. Research has shown that variations in the UTR regions can lead to differences in gene expression levels, which can impact traits such as grain size and yield [59,60]. The UTRs of genes involved in biosynthetic pathways can impact rice grain quality traits, nutritional properties [61,62,63,64,65], and herbicide resistance. Overall, variations in the *γ-TMT* gene underscore the effect of haplotype variation in determining the tocochromanol contents of rice ecotypes.

## 5. Conclusions

This study leveraged resequencing data from 475 rice accessions to delve into the evolutionary aspects and genetic diversity within the *γ-TMT* gene region. We identified a total of 177 genetic variants in the *γ-TMT* region, forming 27 haplotypes. Our genetic diversity analyses highlighted variations among the different rice ecotypes. Phenotypic analysis revealed significant variations in vitamin E profiles, particularly linked to two functional SNPs (fSNPs), *tmt-E2-28,895,665-G/A* and *tmt-E4-28,896,689-A/G*, and a 3’ UTR SNP (*tmt-3UTR-28,897,360-T/A*), impacting vitamin E isomers in both *japonica* and *indica* accessions. Overall, our research provides insights into the genetic diversity and evolutionary dynamics of the *γ-TMT* gene in rice, especially in the context of enhancing its vitamin E content. Through an extensive analysis of various Korean rice accessions, we have pinpointed key genetic variants and haplotypes that significantly influence rice’s nutritional quality. These findings are instrumental for future rice breeding programs aimed at improving the health benefits of this staple crop.

## Figures and Tables

**Figure 1 antioxidants-13-00234-f001:**
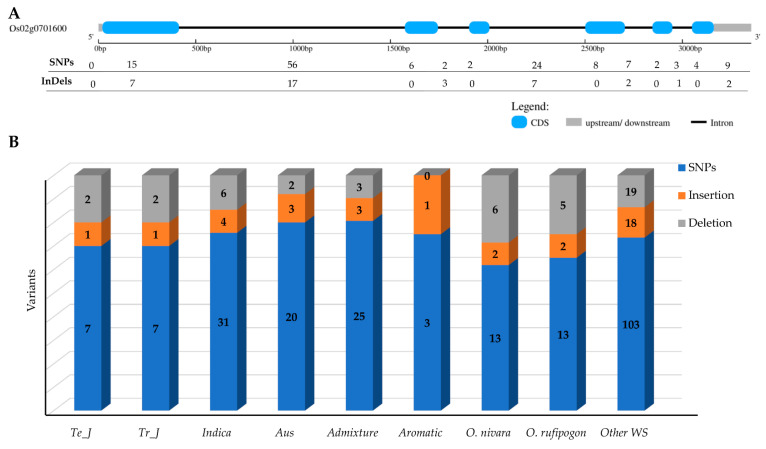
Genetic variants in *γ-TMT* gene region. (**A**) Distribution of SNPs and indels in *γ-TMT* gene region. (**B**) Distribution of SNPs and indels among the ecotypes of KRICE. Te_J: Temperate *japonica*, Tr_J: Tropical *japonica*.

**Figure 2 antioxidants-13-00234-f002:**
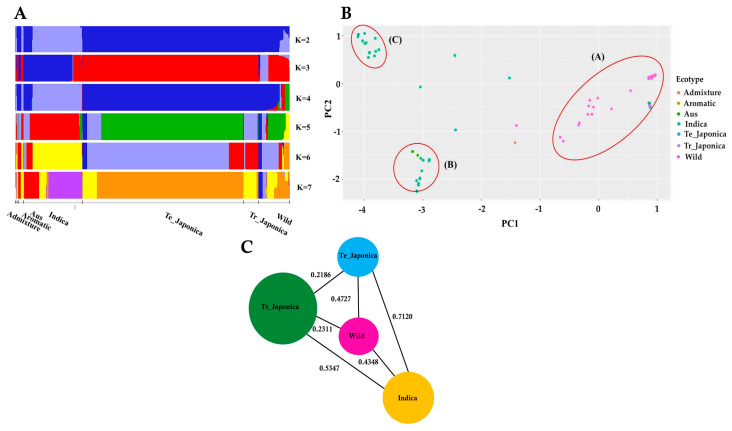
Population structure of KRICE based on the *γ-TMT* gene. (**A**) Population structure of all ecotypes visualized through fastStructure for K-values ranging from 2 to 7, where each color represents a distinct cluster. (**B**) Principal component analysis (PCA) rice accessions. Circle-A: *japonica*, wild, *aromatic*, and some *indica* accessions; Circle-B: *aus* and *indica* accessions; Circle-C: exclusively *indica* accessions. (**C**) Pairwise estimation of genetic differentiation (F*_ST_*) among the ecotypes. Te_: Temperate, Tr_: Tropical.

**Figure 3 antioxidants-13-00234-f003:**
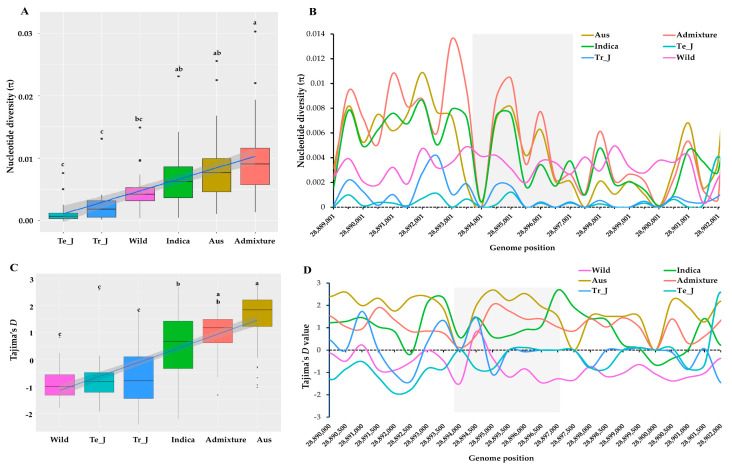
Nucleotide diversity and Tajima’s *D* values based on *γ-TMT* region across ecotypes of KRICE. (**A**) Variation in mean nucleotide diversity among the ecotypes. Different letters above each boxplot indicate significant differences among ecotypes according to Sheffe’s test (*p* < 0.05). (**B**) Nucleotide diversity values with 500 bp sliding window. The highlighted gray color shows the *γ-TMT* gene region. (**C**) Variation in mean Tajima’s *D* values among the ecotypes. Different letters above each boxplot indicate significant differences among ecotypes according to Sheffe’s test (*p* < 0.05). (**D**) Tajima’s *D* values with 500 bp sliding window. The highlighted gray color shows the *γ-TMT* gene region.

**Figure 4 antioxidants-13-00234-f004:**
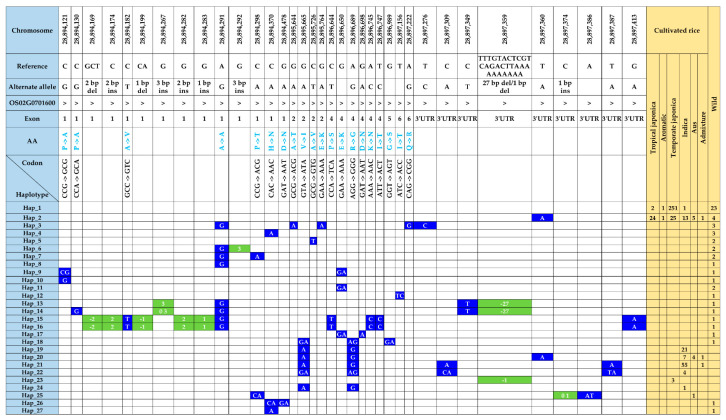
Atlas of haplotype variations in *γ-TMT* transcript region. The distribution of functional SNPs with respective haplotypes is shown with the number of accessions from each ecotype. SNPs and indels are highlighted in blue- and green-colored boxes, respectively, and the blank cell represents the same SNP as the reference.

**Figure 5 antioxidants-13-00234-f005:**
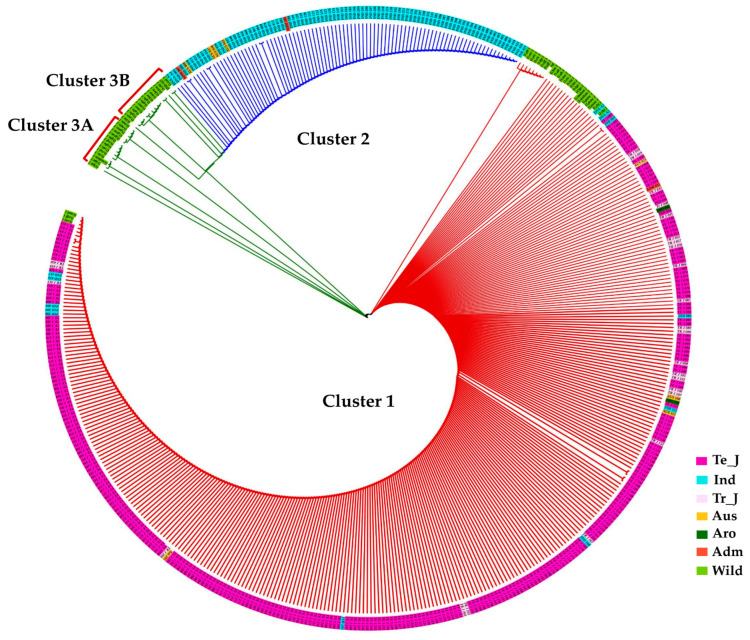
Phylogenetic analysis of KRICE based on *γ-TMT*. Circular dendrogram constructed using the neighbor-joining method with 1000 bootstraps. Te_J: Temperate *japonica*, Tr_J: Tropical *japonica*, Ind: *Indica*, Aro: *Aromatic*, Adm: *Admixture*.

**Figure 6 antioxidants-13-00234-f006:**
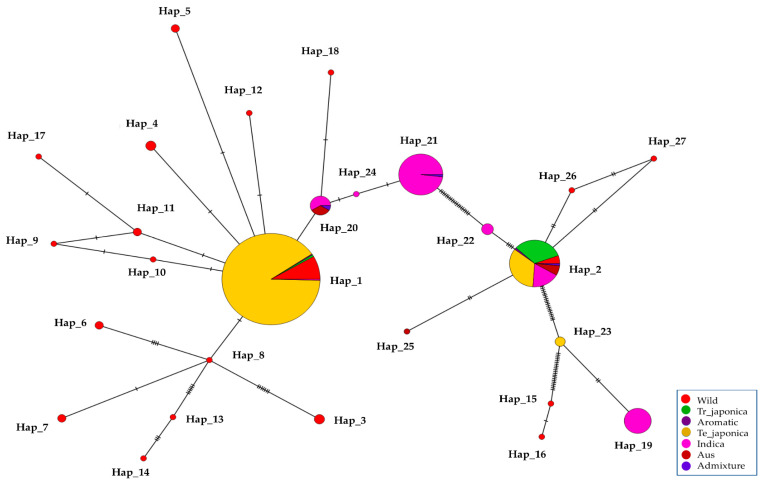
Haplotype network of KRICE based on *γ-TMT* transcript region. The circle size is proportional to the number of samples and ecotypes, and the dashes between haplotypes represent mutational steps between alleles.

**Figure 7 antioxidants-13-00234-f007:**
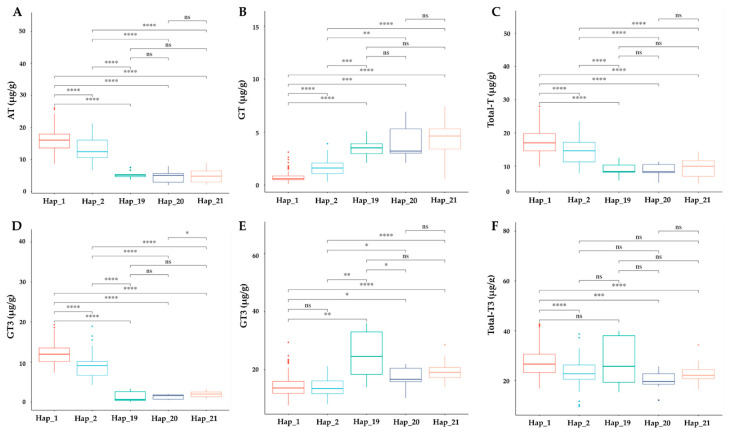
Evaluation of haplotype impact on vitamin E profile. Boxplot showing the impact of haplotypes on (**A**) AT, *α*-tocopherol; (**B**) GT, *γ*-tocopherol; (**C**) Total_T, total tocopherol; (**D**) AT3, *α*-tocotrienol; (**E**) GT3, *γ*-tocotrienol; (**F**) Total_T3, total tocotrienol. The significant difference between each haplotype was investigated with *p* < 0.0001, 0.001, 0.01, 0.05 indicated by ****, ***, **, * and ns (non-significant) based on *t*-test statistics.

## Data Availability

Data are contained within the article and Appendix A.

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
