# Peer review of "Harnessing γ-TMT Genetic Variations and Haplotypes for Vitamin E Diversity in the Korean Rice Collection"

_antioxidants, 2024, doi:10.3390/antiox13020234_

Round 1
Reviewer 1 Report
Comments and Suggestions for Authors
Though plenty of valuable information were provided by this study, we still believe necessary modifications will promote the quality of this manuscript. Firsltly, innvoative results were concluded from the selected sequenced data, while the significant haplotypes lacked of sufficent experimental data to prove. Secondly, the refernces were not uniform, such as some ones lack of issue number, or the published data. Lastly, did the authors chosed the y-TMT at first and then added the relative re-sequencing data or conducted the re-sequencing analysis and then chosed the y-TMT ?
Comments on the Quality of English LanguageThe writing of the manuscript needs improvement, and some grammatical and spelling errors still exist. Suggest to invite someone native to help with the polish.
Author Response
Reviewer 1
Though plenty of valuable information were provided by this study, we still believe necessary modifications will promote the quality of this manuscript.
Response: Thank you for your constructive feedback. We appreciate the opportunity to enhance the quality of our manuscript further.
1. Innovative results were concluded from the selected sequenced data, while the significant haplotypes lacked of sufficient experimental data to prove.
Response: Our study is structured in two main sections: the first part focuses on analyzing genetic diversity to elucidate the variation within the KRICE population. The second part is dedicated to the assessment of vitamin E profiles across major haplotypes (those with more than 5 accessions), covering a total of 240 accessions. We believe that the number of accessions included for haplotype-phenotype association is acceptable to validate haplotypes and their phenotypic effect, particularly in terms of vitamin E profiles.
2. The references were not uniform, such as some ones lack of issue numbers, or the published data. Response: We have thoroughly reviewed and revised all references to ensure uniformity and adherence to the journal's formatting guidelines.
3. Did the authors choose the y-TMT at first and then add the relative re-sequencing data or conduct the re-sequencing analysis and then choose the y-TMT ?
Response: Firstly, we conducted the whole genome re-sequencing of the rice accessions. Following this, we utilized the resequencing data specifically to focus on the haplotype and evolutionary analysis of the γ-TMT gene.
4. The writing of the manuscript needs improvement, and some grammatical and spelling errors still exist. Suggest to invite someone native to help with the polish.
Response: We have thoroughly revised the manuscript for grammatical and spelling accuracy and modified it accordingly.
Reviewer 2 Report
Comments and Suggestions for Authors
The analyzed the γ-TMT gene in 475 Korean rice accessions, uncovering 177 13 genetic variations, they also highlights the genetic landscape of γ-TMT and provides a valuable genetic resource for haplotype-based breeding programs aimed at enhancing nutritional profiles. The manuscript can be published after modification.
Figure 2, 3, 4, 8 are not clear;
α, β, γ and other words should be italics;
Reorganize Figures 2 and 7 if possible;
Conclusions need to be reduced;
Please reorganize the citations according to the initial requirements of the journal
Author Response
Reviewer 2
The analyzed the γ-TMT gene in 475 Korean rice accessions, uncovering 177 13 genetic variations, they also highlights the genetic landscape of γ-TMT and provide a valuable genetic resource for haplotype-based breeding programs aimed at enhancing nutritional profiles. The manuscript can be published after modification.
Response: Thank you for your valuable suggestions. We have modified the manuscript as per the suggestions.
1. Figures 2, 3, 4, and 8 are not clear.
Response: We have adjusted the figure resolution to be clear as suggested. Actually, all figure are in >300dpi, resolution might be getting reduced during PDF conversion. Therefore, we have also attached zip file for all figures in TIF format.
2. α, β, γ and other words should be in italics
Response: We have corrected it in the revised manuscript.
3. Reorganize Figures 2 and 7 if possible.
Response: As suggested, we have reorganized figure no. 2 and 7.
4. Conclusions need to be reduced
Response: Thank you very much for pointing this out. We have trimmed the conclusion in revised manuscript.
5. Please reorganize the citations according to the initial requirements of the journal
Response: We have thoroughly reviewed and revised all references to ensure uniformity and adherence to the journal's formatting guidelines.
Reviewer 3 Report
Comments and Suggestions for Authors
The submitted manuscript reports detailed sequence-level analysis of the Gamma-tocopherol methyltransferase (γ-TMT) 475 rice accessions by means of of short-read mapping of 15x Illumina resequencing. The manuscript represents well-focused and correctly implemented research. The paper is well structured and it reads well. I have no major concerns about the content of the manuscript. Below are mine remarks and suggestions:
Line 14:
"genetic variations" change to "genetic variants" (at any further occurrences as well).
The correct collecting term for sequence-level genetic differences (like SNPs, InDels, substitutions, etc) is "genetic variants" and not "variations".
The author used this incorrect term at many places and they should be corrected.
N.B.:
In other contexts, like "To elucidate the composition of KRICE in terms of shared genetic variation in γ-TMT", (Line 211), using "genetic variation" might be correct.
Lines 118-119:
"To eliminate missing values we used VCFtools..."
I assume that "eliminating missing values" means imputing in this case. In any case, the author should give more information about the procedure and its parameters.
Page 8 -circular dendogram:
In the main text it's indicated that the accessions are grouped into three main cluster. However, the figure is somewhat equivocal at the first glance, as cluster 2 (blue) interrupts two, predominantly green segments. It would probably be beneficial to show the exact cluster boundaries on an additional outer circle (or ring).
The paper is well structured and it reads well.
Author Response
Reviewer 3
The submitted manuscript reports detailed sequence-level analysis of the Gamma-tocopherol methyltransferase (γ-TMT) 475 rice accessions by means of of short-read mapping of 15x Illumina resequencing. The manuscript represents well-focused and correctly implemented research. The paper is well structured and it reads well. I have no major concerns about the content of the manuscript. Below are mine remarks and suggestions:
Response: Thank you very much for your appreciation and thorough review of our manuscript. We have addressed the comments mentioned here and tracked in the revised manuscript.
1. Line 14: "genetic variations" change to "genetic variants" (at any further occurrences as well). The correct collecting term for sequence-level genetic differences (like SNPs, InDels, substitutions, etc) is "genetic variants" and not "variations". The author used this incorrect term at many places and they should be corrected. N.B.:In other contexts, like "To elucidate the composition of KRICE in terms of shared genetic variation in γ-TMT", (Line 211), using "genetic variation" might be correct.
Response: We have changed "genetic variations" to "genetic variants" as suggested.
2. Lines 118-119:
"To eliminate missing values we used VCFtools..."I assume that "eliminating missing values" means imputing in this case. In any case, the author should give more information about the procedure and its parameters.
Response: Thanks. We have provided more information on this point in revised manuscript.
3. Page 8 -circular dendrogram: In the main text it's indicated that the accessions are grouped into three main cluster. However, the figure is somewhat equivocal at the first glance, as cluster 2 (blue) interrupts two, predominantly green segments. It would probably be beneficial to show the exact cluster boundaries on an additional outer circle (or ring).
Response: We have modified the dendrogram as suggested.
Round 2
Reviewer 2 Report
Comments and Suggestions for Authors
The quality of the manuscript has been improved as a whole after the revision, and the paper can be published.
Adding Latin scientific names such as rice, soybean, tomato, maize after the plant name.
The font in Figure 2 is too small, please enlarge it;
It is suggested that Figures 3 and 4 should be merged to reduce the space for publication

Fine.
Author Response
Reviewer 2
The quality of the manuscript has been improved as a whole after the revision, and the paper can be published.
Response: Thank you very much for the appreciation and valuable suggestions for further improvement of our manuscript.
1. Adding Latin scientific names such as rice, soybean, tomato, maize after the plant name.
Response: Thanks. We have provided Latin scientific names of crops at their first appearance.
2. The font in Figure 2 is too small, please enlarge it
Response: We have adjusted the font size accordingly. Actually, all figures are with 400dpi resolution, it might be they are getting reduced during PDF conversion. Therefore, we have also submitted all figures in TIF format separately in the system.
3. It is suggested that Figures 3 and 4 should be merged to reduce the space for publication
Response: As per the reviewer's suggestion, we have merged Figures 3 and 4.